# Biological Evaluation of *Valeriana* Extracts from Argentina with Potent Cholinesterase Inhibition for the Treatment of Neurodegenerative Disorders and Their Comorbidities—The Case of *Valeriana carnosa* Sm. (Caprifoliaceae) Studied in Mice

**DOI:** 10.3390/ph16010129

**Published:** 2023-01-16

**Authors:** Carolina Marcucci, Marina Rademacher, Fabiola Kamecki, Valentina Pastore, Hernán Gerónimo Bach, Rafael Alejandro Ricco, Marcelo Luis Wagner, Damijan Knez, Stanislav Gobec, Natalia Colettis, Mariel Marder

**Affiliations:** 1Facultad de Farmacia y Bioquímica, Instituto de Química y Fisicoquímica Biológicas Prof. Dr. Alejandro C. Paladini, Consejo Nacional de Investigaciones Científicas y Técnicas, Universidad de Buenos Aires, Junín 956 (C1113AAD), Buenos Aires C1053ABH, Argentina; 2Departamento de Farmacología, Facultad de Farmacia y Bioquímica, Universidad de Buenos Aires, Cátedra de Farmacobotánica, Junín 956 (C1113AAD), Buenos Aires C1053ABH, Argentina; 3Instituto Nacional de Tecnología Agropecuaria (INTA), Instituto de Recursos Biológicos, De Los Reseros y Nicolas Repetto s/n (1686) Hurlingham, Buenos Aires C1033AAE, Argentina; 4Faculty of Pharmacy, University of Ljubljana, Aškerčeva 7, 1000 Ljubljana, Slovenia

**Keywords:** Alzheimer’s disease, medicinal plant, acetylcholinesterase, antioxidant, β-amyloid aggregation, monoamine oxidase, memory, antidepressant

## Abstract

Alzheimer’s disease (AD) is a neurodegenerative disorder whose pathophysiology includes the abnormal accumulation of proteins (e.g., β-amyloid), oxidative stress, and alterations in neurotransmitter levels, mainly acetylcholine. Here we present a comparative study of the effect of extracts obtained from endemic Argentinian species of valerians, namely *V. carnosa* Sm., *V. clarionifolia* Phil. and *V. macrorhiza* Poepp. ex DC from Patagonia and *V. ferax* (Griseb.) Höck and *V. effusa* Griseb., on different AD-related biological targets. Of these anxiolytic, sedative and sleep-inducing valerians, *V. carnosa* proved the most promising and was assayed *in vivo*. All valerians inhibited acetylcholinesterase (IC_50_ between 1.08–12.69 mg/mL) and butyrylcholinesterase (IC_50_ between 0.0019–1.46 mg/mL). They also inhibited the aggregation of β-amyloid peptide, were able to chelate Fe^2+^ ions, and exhibited a direct relationship between antioxidant capacity and phenolic content. Moreover, *V. carnosa* was able to inhibit human monoamine oxidase A (IC_50_: 0.286 mg/mL (0.213–0.384)). A daily intake of aqueous *V. carnosa* extract by male Swiss mice (50 and 150 mg/kg/day) resulted in anxiolytic and antidepressant-like behavior and improved spatial memory. In addition, decreased AChE activity and oxidative stress markers were observed in treated mouse brains. Our studies contribute to the development of indigenous herbal medicines as therapeutic agents for AD.

## 1. Introduction

Valerian is the common name for a perennial herb plant belonging to the family Caprifoliaceae and native to various regions of America, Europe and Asia [1]. Among approximately 200 species known so far, *Valeriana officinalis* L. is the one most commonly used for medicinal purposes [1]. Reports have shown that some valerian species have been used for centuries in Europe and North America as mild sedatives and sleep enhancers [2,3]. Analyzing ancient writings and modern pharmacological research on the *Valeriana* genus, we have reported pharmacological activity including anxiolytic, antidepressant, antispasmodic, sedative, antitumor and anti-HIV properties, as well as its use in the treatment of neurological diseases [4,5,6,7,8]. Previous phytochemical studies on this species led to the isolation of flavonoids [9,10,11], essential oils [12], sesquiterpenoids [13], valepotriates and acylated iridoids [13,14].

About 81 species of the *Valeriana* genus are grown in South America, 48 of which have been found in Argentina [15,16], mainly in the Andes. Although some of these species are used in folk medicine, pharmacological, phytochemical and botanical information about most of them, and their quality and efficacy as phytopharmaceuticals or dietary supplements, remains scarce. Therefore, we conducted a comparative study on the phytochemical composition and potential central nervous system (CNS) effects of hydroalcoholic extracts of a series of Argentine valerians including *Valeriana carnosa* Sm., *V. clarionifolia* Phil. and *V. macrorhiza* Poepp. ex DC. from Argentina Patagonia; *V. ferax* (Griseb.) Höck and *V. effusa* Griseb. from the central region of Argentina, and *V. officinalis* as a reference plant. All these plants evidenced a high content of phenolic compounds and presented ligands for the benzodiazepine binding site of the GABA_A_ receptor, showing sedative effects (500 mg/kg, intraperitoneally, i.p.) in mice in the sodium thiopental-induced loss of righting reflex assay. In addition, extracts of *V. macrorhiza*, *V. carnosa* and *V. ferax* induced a decrease in exploratory behavior, whereas *V. clarionifolia* produced anxiolytic-like activity (500 mg/kg, i.p.) in the hole board test. Acute peroral treatment (300 mg/kg and 600 mg/kg, p.o.) showed sedative activity for extracts of *V. ferax* and anxiolytic properties for extracts of *V. macrorhiza*, *V. carnosa* and *V. clarionifolia*. These native species were thus active in the CNS, and their use as anxiolytics/sedatives and sleep enhancers in folk medicine was validated [10].

Alzheimer’s disease (AD) is a neurodegenerative disorder characterized by progressive neuronal degeneration, including personality changes, dysmnesia, emotional disturbances and behavioral abnormalities [17]. AD etiology is not yet fully understood, although some mechanisms such as oxidative stress [18], low levels of neurotransmitter acetylcholine (ACh) [19] and the aggregation and accumulation of β-amyloid (Aβ) plaques [20] are thought to play an important pathogenic role. ACh can be hydrolyzed by the two cholinesterases (ChEs), i.e., acetylcholinesterase (AChE) and butyrylcholinesterase (BChE). AChE selectively hydrolyzes ACh, while BChE can metabolize several neuroactive neurotransmitters and peptides, including ACh and acylated ghreline [21]. Cognitive improvements in AD patients have been associated with higher levels of BChE inhibition [22], particularly in spatial, verbal, memory, speed and reaction time tasks. Drugs inhibiting AChE and/or BChE such as rivastigmine, donepezil, and galantamine are clinically used to treat AD [23].

Polypharmacology is the development and use of therapeutics acting on multiple targets of a disease and/or its comorbidities. Indeed, multitarget drugs acting in an interrelated network offer potentially higher efficacy and may curve the drawbacks associated with the use of single-target drugs [24]. A multitarget drug may be a single compound with multiple activities, a mixture of compounds with single targets, or even a mixture such as a plant extract. Considering the multifactorial etiology of AD and the number of potential therapeutic targets, new therapeutic options should be based on a multitarget approach [25]. Current therapies acting on a single target provide temporary relief but fail to prevent the progression of AD or slow down neurodegeneration [26,27]. Consequently, strategies counteracting Aβ aggregation and restoring neurotransmitter levels through combined inhibition of ChEs and monoamine oxidases (MAOs) may represent a therapeutically feasible approach. MAOs catalyze the oxidative deamination of monoamines such as serotonin (MAO-A), norepinephrine and dopamine (MAO-B), also generating hydrogen peroxide, which increases oxidative stress.

Among natural neuroactive products, attention has been drawn to the anxiolytic properties of the *Valeriana* genus [28]. *V. officinalis* [7], *V. prionophylla* [29] and *V. wallichii* [30,31] are not only the most popular herbal medicines in the treatment of anxiety and mild sleep disorders, but have also shown antidepressant-like activity in some preclinical studies. Extracts from valerian have also inhibited AChE [32,33], and a possible AD-related therapeutic effect of *V. amurensis* has been reported [34,35]. Moreover, *V. carnosa*, better known as “*ñamkulawen*”, is one of the five most best-known native species from Argentine Patagonia in the pharmacopoeia of the local communities and may be the most popular species with medicinal properties [36]. *V. carnosa* is locally thought to have “panacea-like properties”, which gives it a high cultural and symbolic value for the Mapuche community. Furthermore, the reputation and use of *V. carnosa* have spread to the formal and informal market for medicinal herbs in urban Patagonia [37].

In this paper, we present a comparative study of the effects of Argentine valerians on targets related to AD and a more comprehensive *in vivo* study of the most promising Argentine *Valeriana* species—*V. carnosa*.

## 2. Results and Discussion

### 2.1. Plant Material and Extraction

The studies were carried out on roots and rhizomes of *Valeriana carnosa* Sm., *V. clarionifolia* Phil. and *V. macrorhiza* Poepp. ex DC., collected in Argentine Patagonia; and *V. ferax* (Griseb.) Höck and *V. effusa* Griseb collected in the center of the country. The collection of plant material was in accordance with national Argentine guidelines. All plants were authenticated by Dr. Hernán Bach and voucher specimens were deposited in the herbarium of the Institute of Biological Resources -INTA and in the “Juan Anibal Domínguez” herbarium, Museum of Pharmacobotany, as previously reported [10] (Appendix A). Fresh roots and rhizomes were washed, dried at room temperature, and protected from light. Part of these roots and rhizomes was then pulverized, and the resulting powders were subjected to the extraction and fractionation scheme shown in Figure 1. Powdered dry roots and rhizomes (50 g) of each plant were suspended in 500 mL of 70% ethanol, and the mixture was kept at 25 °C for 2.5 h with shaking. The filtrate was concentrated to 1/3 of the original volume to remove most of the ethanol and extracted with an equal volume of petroleum ether, which was discarded. Two-thirds of the aqueous phase obtained was evaporated to dryness (aqueous 1 extract) and used for chemical, biochemical, and pharmacological studies. The other third was slightly concentrated to remove the remaining petroleum ether and extracted twice with an equal amount of ethyl ether. The two resulting phases were evaporated to dryness (aqueous 2 and ethylic extracts), and the residues were stored protected from day-light at –20 °C. Replicates of the same batch were evaluated.

### 2.2. Antioxidant Effect of Aqueous 1 Extracts

The results obtained for aqueous 1 extracts of *Valeriana effusa*, *V. ferax*, *V. macrorhiza*, *V. clarionifolia*, *V. carnosa* and *V. officinalis* on the 2,2-diphenyl-1-picrylhydrazyl (DPPH), 2,2߰-azino-bis(3-ethylbenzothiazoline)-6-sulfonic acid (ABTS), iron chelation, and thiobarbituric acid (TBA) reactive substances (TBARS) assays are shown in Table 1. In addition, the total content of polyphenols (obtained from [10]) is also reported for comparison and discussion. All species showed good antioxidant capacity against DPPH (EC_50_ of 0.12 ± 0.02 mg/mL for *V. carnosa* to 0.57 ± 0.02 mg/mL for *V. ferax*) and ABTS radicals (EC_50_ of 0.04 ± 0.01 mg/mL for *V. carnosa* to 0.51 ± 0.06 mg/mL for *V. officinalis*). This effect may be due to the presence of phenolic compounds in these plant extracts. Indeed, a direct relationship was observed between the amount of polyphenols and the extracts’ antioxidant capacity (content of polyphenols of 191.56 ± 8.99 mg galic acid/1 g plant for *V. ferax* to 890.25 ± 156.05 mg galic acid/1 g plant for *V. carnosa*). As phenolic compounds exhibit antioxidant activity by mechanisms such as hydrogen atom donation to free radicals and reactive oxygen species (ROS) scavenging, they may stabilize DPPH or ABTS radicals by either absorption or deactivation through bonding/coordination [38]. ROS are chemically reactive radicals produced as byproducts of primary metabolism and can trigger serious oxidative damage in macromolecules. Excessive ROS leads to damage in enzymatic processes and alters the function of various lipids, proteins, and DNA, which may result in neurodegenerative diseases, among others. Some phenols can also bind to transition metal ions (especially iron), often resulting in forms poorly active in promoting free radical reactions [39]. As evident in the ferrozine assay, all extracts chelated Fe^2+^ (IC_50_ of 5.13 ± 0.09 mg/mL for *V. macrorhiza* to 1.3 ± 0.11 mg/mL for *V. clarionifolia;* close to the value obtained for *V. carnosa*, 1.71 ± 0.10 mg/mL). The aqueous 1 extract of *V. carnosa* showed the highest polyphenols concentration and the lowest IC_50_ and EC_50_ values, which makes it the most active extract as a free-radical scavenger and Fe^2+^ chelator (Table 1). In addition, all extracts reduced lipid peroxidation in the *in vitro* TBARS assay, with *V. macrorhiza* proving to be the most active (IC_50_ of 0.20 (0.13–0.30) mg/mL). Worth pointing out, the aqueous 1 extract of *V. carnosa* was not evaluated in the *in vitro* TBARS assay, as it decomposed in the reaction assay. So, its ethylic extract was tested instead, proving to be the most active (IC_50_ of 0.18 (0.11–0.30) mg/mL).

The results shown above agree with those reported for other species of the *Valeriana* genus. The methanolic and aqueous extracts of *Valeriana jatamansi* have shown IC_50_ values of 0.078 mg/mL and 0.154 mg/mL, respectively, in the DPPH assay. However, the essential oil of *V. jatamansi* shows lower free radical scavenging activity (IC_50_ of 0.876 mg/mL), while the chloroform extract shows negligible activity. In addition, a linear correlation has been observed between the antioxidant activity and the content of polyphenols and flavonoids in the extracts [40]. Moreover, the oils of *V. hardwickii* (IC_50_ = 1.10 mg/mL) [41] and *V. officinalis* (IC_50_ of 0.493 mg/mL) [42] show results similar to those of *V. jatamansi* and the antioxidant capacity reported for *V. wallichii* is also directly proportional to its phenolic content, which seems to be the case for many plant species [43]. Aqueous and alcoholic extracts (70%) of *V. wallichii* have been able to scavenge both ABTS and DPPH radicals, with the hydroalcoholic extract being more active than the aqueous one [44]. In turn, the ethanolic extract of *V. alliariifolia* has shown good antioxidant activity against DPPH (IC_50_ 0.018 mg/mL) and ABTS (IC_50_ 0.024 mg/mL), also exhibiting the highest total phenolic content [45]. All these results suggest that compounds with antioxidant activity in many *Valeriana* species have high to moderate polarity. Other *Valeriana* species evidence the ability to chelate Fe^2+^ with *V. officinalis* oil showing an IC_50_ of 0.235 mg/mL [46]. Additionally, the methanolic extract of *V. jatamansi* has demonstrated good chelation activity (76%), followed by aqueous extracts (43%) and essential oils (31%) at a concentration of 0.1 mg/mL. In contrast, chloroform extracts show poor chelation activity (12%) [40].

### 2.3. Inhibitory Activity of Extracts against AChE 

The results obtained for aqueous 1 and ethylic extracts of valerians in murine recombinant AChE (mAChE) assays and mouse brain homogenates are shown in Table 2.

Ethylic extacts—though not aqueous 1 extracts— inhibited recombinant mAChE at 1 mg/mL. This result suggests that rather nonpolar compounds are responsible for inhibition and that they are found in small amounts in aqueous extracts. In turn, the aqueous 1 extracts of *V. clarionifolia* and *V. macrorhiza* were the most potent AChE inhibitors in mouse brain homogenates, while the ethylic extracts of *V. clarionifolia* and *V. carnosa* inhibited mAChE. Tacrine, the first centrally acting inhibitor approved for the treatment of AD, was chosen as the reference compound.

The inhibitory activity of other species in the *Valeriana* genus against ChEs has been scarcely documented. Chloroform and ethyl acetate extracts of *Valeriana wallichii* significantly inhibit AChE (IC_50_ 0.061 mg/mL) and BChE (IC_50_ 0.058 mg/mL), while extracts from *V. polystachya* also show slight AChE inhibition at 200 µg/mL. In addition, compounds 6β-hydroxysitostenone, baldrine, and IVHD-valtrate isolated from *V. polystachya* inhibit AChE at 150 μM (53.8, 37.3 and 38.4% inhibition, respectively) [47]. Moreover, studies with iridoids and sesquiterpenoids isolated from the ethyl acetate extract of *V. jatamansi*, which have revealed weak inhibition of AChE, showed that these compounds were unable to inhibit mAChE at 50 µM and therefore not responsible for total extract activity [48]. On the other hand, some compounds isolated from the ethanolic extract of *V. officinalis*—i.e., spatulenol (49.1%), anismol A (20.2%), (þ)-8-hydroxypinoresinol (19.3%), and pinoresinol-4-*O*-β-*D*-glucopyranoside (32.0%) have shown moderate activity on mAChE at 100 µm [33]. For these reasons, extracts from Argentine species show good inhibitory ability against AChE and similar to other species of this plant.

### 2.4. Inhibitory Activity of the Extracts against BChE

The results obtained for aqueous and ethylic extracts of valerians in human recombinant BChE (hBChE) and mouse plasma BChE inhibition assays are shown in Table 2.

Aqueous 1 extracts were able to inhibit both human and murine BChE, with more potent inhibition of the murine enzyme. However, comparisons should be made with caution, as reaction conditions and media differed between assays. Nevertheless, the extracts of *V. clarionifolia* were the most active, and ethylic extracts were more potent BChE inhibitors than aqueous extracts 1 and 2. This result indicates that nonpolar compounds are responsible for overall activity, as they are extractable with a nonpolar solvent such as ethyl ether.

We found no reports in the literature describing valerians such as BChE inhibitors, besides *V. wallichii,* although inhibition by other species has certainly been described. Studies on ChE inhibition by methanol extracts of different parts of *Nelumbo nucifera* have shown weak ChE inhibition by rhizomes, potent BChE inhibition by embryos and active AChE inhibition by seeds [49,50]. *Artemisia scoparia* and *Artemisia absinthium* leaves’ essential oils also show anticholinesterase potential by inhibiting both AChE (IC_50_ of 30 ± 0.04 µg/mL and 32 ± 0.05 µg/mL, respectively) and BChE (IC_50_ of 34 ± 0.07 µg/mL and 36 ± 0.03 µg/mL, respectively) [51]. Another study has described the inhibitory activity against BChE of 26 ethanolic extracts of Chinese medicinal plants. For 15 of them, underground parts revealed IC_50_ values between 0.005–0.3 mg/mL. Among them, *Polygonum multiflorum* was the most potent BChE inhibitor (IC_50_ = 0.005 mg/mL), while *Paeonia lactiflora* and *Paeonia veitchii* also potently inhibited BChE with IC_50_ values of 0.006 and 0.01 mg/mL, respectively. Interestingly, some extracts of *Angelica dahurica*, *Cremastra appendiculata*, *Juncus setchuensis* and *Nardostachys jatamansi* selectively inhibited BChE in comparison to AChE [52]. Of note, *Nardostachys jatamansi* belongs to the same family as the *Valeriana* genus and inhibits BChE with an IC_50_ of 0.015 mg/mL. To sum up, the extracts presented here are potent BChE inhibitors when compared to extracts reported in the literature.

The BChE/AChE ratio changes considerably in cortical regions affected by AD [53]. In fact, in early stages of AD AChE levels decrease up to 85% in some brain regions, whereas BChE levels increase with disease progression [53]. While increased BChE activity aggravates AD, selective BChE inhibition ameliorates the cholinergic deficit. Notably, selective BChE inhibition has been shown to elevate extracellular ACh levels and to improve learning in elderly rats performing maze trials [54]. Adverse effects may be associated with a lack of central vs. peripheral selectivity (e.g., muscle cramps) or brain regional selectivity (e.g., sleep disturbances and extrapyramidal symptoms), in turn, due to poor inhibitor preference for AChE or BChE. Worth highlighting, the plants used in this work inhibit BChE to a greater extent than AChE, which could prove a beneficial feature associated with fewer side effects.

### 2.5. Inhibitory Activity of Extracts against MAO

The results obtained for aqueous 1 and ethylic extracts in human MAO-A (hMAO-A) and hMAO-B inhibition assays are shown in Table 3.

*V. officinalis* aqueous 1 extract and *V. carnosa* and *V. macrorhiza* ethylic extracts were the most potent hMAO-A inhibitors (the IC_50_ value was calculated for *V. carnosa*). Moreover, ethylic extracts showed greater inhibitory potency than aqueous 1 extracts, except for *V. clarionifolia*. Again, this suggests that compounds responsible for total activity are rather lipophilic. On the other hand, the aqueous 1 and ethylic extracts of *V. carnosa*, *V. macrorhiza*, and *V. officinalis* had greater ability to inhibit hMAO-B. In this case, aqueous 1 extracts were more active than ethylic extracts which makes polar compounds responsible for inhibitory effects.

MAO-A inhibitors were the first antidepressants that appeared on the market. Clorgyline is an irreversible and selective inhibitor of MAO-A used in research and here as a reference. Unlike MAO-A, MAO-B is associated with inhibition of dopamine degradation, rather than that of noradrenaline and serotonin metabolism, [55]. Therefore, MAO-B inhibitors are used to slow down the progression of Parkinson’s disease. Pargyline is an irreversible MAO-B inhibitor that has been used to treat depression, whereas *L*-deprenyl (selegiline) is an antiparkinsonian and antidepressant used to treat early stages of Parkinson’s disease, depression, and senile dementia. Selegilin selectively and irreversibly inhibits MAO-B, but loses specificity at higher doses, thus inhibiting MAO-A [56]. In turn, isatin (indole-2,3-dione) is an endogenous inhibitor of MAO B [57]. Furthermore, MAO inhibitors inhibit hydrogen peroxide production, which triggers a neuroprotective effect [58].

To the best of our knowledge, this is the first report linking an extract of the *Valeriana* genus with MAO inhibition. Other medicinal plant species with antidepressant activity have been reported as MAO inhibitors, for instance, *Hypericum perforatum*, has been shown to inhibit MAO-A and MAO-B (0.005 mg/mL, 93% and 49%, respectively) [59]. The methanolic extracts of commercial *H. perforatum* powder (Herb Pharm, Williams, OR, USA) inhibit hMAO-A with an IC_50_ of 0.142 ± 0.031 mg/mL, whereas the methanolic flower extracts of *H. perforatum* show the most potent inhibition (IC_50_ of 0.0636 ± 0.0094 mg/mL), followed by stems and leaves (IC_50_ of 0.1436 ± 0.0165 mg/mL), and finally by the root extracts. In addition, the ethylic extract of the underground parts of *V. carnosa* showed better inhibitory capacity (IC_50_ (mg/mL) of 0.286 (0.213–0.384)) than the root extracts of this widely used antidepressant medicinal plant. Another species with MAO inhibition capacity is *Peganum harmala*, which seed extracts strongly inhibit hMAO-A (IC_50_ of 0.0499 ± 0.0056 mg/L) [60]. Of note, underground part extracts of the Argentine *Valeriana* species show MAO inhibitory effects comparable to those of other species used as antidepressants in traditional medicine and clinics such as *H. perforatum*. 

### 2.6. Inhibitory Activity of Extracts against Aβ_1-42_ Aggregation

All aqueous 1 extracts of *Valeriana* species were able to inhibit Aβ aggregation at 0.1 mg/mL (Table 4). Some small molecules such as resveratrol, curcumin, and their derivatives, have exhibited neuroprotective effects by inhibiting Aβ-induced damage, inhibiting ChEs and attenuating neuronal death by oxidative stress [40,61,62]. The aromatic structures of these compounds contribute to the prevention of self-induced Aβ aggregation by disrupting the π stacking of aromatic residues in Aβ aggregate development [63]. In particular, resveratrol (*trans*-3,4′, 5-trihydroxystilbene), a flavonoid derivative largely found in grapes [64], inhibits Aβ self-aggregation, attenuates Aβ induced toxicity, promotes Aβ clearance, and reduces senile plaques [65]; for these reasons this compound was used as a positive control in Aβ_1-42_ aggregation inhibition assays. The results shown in Table 4 reveal that *V. effusa* and *V. clarionifolia* were the most potent inhibitors, with inhibition values similar to those of resveratrol. 

Few reports have described the effect of plant extracts on Aβ aggregation and, to our knowledge, none have described this effect for the *Valeriana* genus. A study has reported 58.3 ± 1.0 % Aβ aggregation inhibition for the aqueous extract of *Lycium barbarum* fruits at 0.1 mg/mL [66]. In addition, analyses of 14 aqueous extracts of plant seeds (68–123 mg/mL) revealed inhibition values of 98.7 ± 2.4% for lettuce, 95.9 ± 2.6% for bitter melon, 93.9 ± 2.1% for corn, and 68.3 ± 1.9% for the common daisy [67]. Another study reported the effect of ethanolic extracts of Chinese medicinal plants, where the underground parts were studied for 15 out of total 26 plants, with inhibition values ranging from 10 to 92% (at 0.1 mg/mL). *Rheum officinale* showed promising inhibitory effects (91.75%) as compared to curcumin (positive control) (92.97%), followed by *Polygonum multiflorum* root (87.41%), and *Cremastra appendiculata* and *Paeonia veitchii* (under 73%) [52]. 

Briefly, Argentine valerian extracts exhibited moderate to high capacity to inhibit Aβ aggregation. But this is only one among other properties described in this and previous papers [10,68,69]. All these characteristics make Argentine valerian extracts multitarget drugs for the treatment of neurodegenerative diseases and their comorbidities.

The aqueous 1 extract of *V. carnosa* inhibited both AChE (IC_50_ 6.71 (2.86–15.77) mg/mL) and BChE (IC_50_ 1.46 (0.99–1.14) mg/mL). In addition, it inhibited Aβ aggregation, was able to chelate Fe^2+^ and in having the highest polyphenol content, was the most potent free radical scavenger. Moreover, *V. carnosa* was able to inhibit hMAO-A (IC_50_: 0.286 (0.213–0.384) mg/mL) and hMAO-B. Although not the most active extract in all the targets studied aqueous 1 of *V. carnosa* was an active extract in all targets. For these reasons, *V. carnosa* proved to be the most promising species within this group of Argentine valerians and was selected for the *in vivo* studies in mice.

### 2.7. Effect of Chronic Treatments of V. Carnosa Aqueous 1 Extract in Mice 

#### 2.7.1. Stability Determination by High Performance Liquid Chromatography (HPLC) Fractionation

An aliquot of the extract solution was chromatographed directly after extraction, after frozen sample reconstitution to be used for *in vivo* study and after it was consumed by mice (before it was renewed). No significant differences were observed in the chromatograms obtained after extraction, before or after the animal intake (Appendix A). Two major constituents had already been determined in the aqueous 1 extract of *V. carnosa*, [10], *2S*-hesperidin and chlorogenic acid (~1.5% and 12.0%, respectively). 

*2S*-Hesperidin (Appendix A) is a glycosylated flavanone with sedative-hypnotic effects [11]. Chronic oral ingestion of this flavonoid elicits anxiolytic-like effects. *2S*-Hesperidin inhibits AChE (IC_50_ = 0.85 nM) and BChE (IC_50_ = 3.45 nM) [70]. On the other hand, hesperidin shows relatively weak hMAO inhibition [71], but its chronic treatment in rats (100 mg/kg) inhibits MAO-A hippocampal activity [72]. Moreover, 10 mM hesperidin can inhibit ~40% of Aβ formation *in vitro* [73]. In addition, hesperidin can scavenge free radicals DPPH (247.52 µg/mL) [74] and ABTS (16.46 µM) [73]. Oral administration of hesperidin to rats (50 mg/kg/day) for 10 consecutive days before and 14 days after γ-irradiation significantly attenuates radiation-induced TBARS levels [75]. In addition, chronic treatment with hesperidin (1 mg/kg) produces antidepressant-like effects in mice in the tail suspension test [76]. In Wistar rats, hesperidin (25, 50, and 100 mg/kg) restores alternation index levels decreased by L-methionine in the Y-maze test. In addition, these rats exhibits decreased AChE activity and malondialdehyde (MDA) concentration, and increased glutathione (GSH) levels [77].

In turn, polyphenol chlorogenic acid is an ester of caffeic acid and L-quinic acid (Appendix A) which inhibits AChE (IC_50_ of 8.01 µg/mL), BChE (IC_50_ of 6.3 µg/mL) [78], and hMAO-A (IC_50_ of 158 µM) but fails to inhibit hMAO-B at maximum concentration tested (i.e., 100 µM) [79]. Administration of chlorogenic acid (6 or 9 mg/kg) significantly restores the alternation index previously decreased by scopolamine in mice. Chlorogenic acid at doses of 3, 6, and 9 mg/kg significantly inhibits AChE activity in the hippocampus, and reduces hippocampal MDA levels in scopolamine treated mice at 3 or 6 mg/kg doses [80]. In addition, chlorogenic acid scavenges free radicals such as DPPH (IC_50_ of 35 μg/mL) and ABTS (IC_50_ of 38 μg/mL) [81]. Chlorogenic acid-enriched extracts of *Eucommia ulmoides* (200 and 400 mg/kg/day for 7 days) have shown antidepressant-like activity in mice in the tail suspension test. Moreover, chlorogenic acid has been detected in high-performance liquid chromatography-electrospray ionization-tandem mass spectrometry (UHPLC-ESI-MS/MS) analysis of cerebrospinal fluid of rats treated with oral chlorogenic acid-enriched extracts of *E. ulmoides*. This finding indicates that chlorogenic acid can cross the blood-brain barrier in rodents [82]. 

In sum, the two compounds present in the aqueous 1 extract of *V. carnosa* are active in the CNS. Consequently, extract activities could be due to each compound, the possible pharmacological interactions between them (which may be synergistic) or pharmacological interaction with other compounds. Thus, the biological results for this extract may thus result from a complex combination of activities of the individual compounds and/or a combination of interactions between them.

#### 2.7.2. Fluid Intake and Mouse Weight

The effects of *V. carnosa* aqueous 1 extract intake (50 and 150 mg/kg/day, groups 2 and 3, respectively) on the amount of fluid ingested and the mouse weight are shown in Appendix A. Weight was monitored to assess possible adverse effects such as loss of appetite or decreased mobility. On the other hand, the amount of daily fluid intake allowed us to determine doses and assess whether mice avoided ingestion because of organoleptically unpleasant fluid properties. We found no differences in weight or fluid intake between the control group that drank water (group 1) and the treated groups (groups 2 and 3).

#### 2.7.3. Effect of *V. carnosa* in the Hole Board and Locomotor Activity Assays

The effects of chronic ingestion of the aqueous 1 extract of *V. carnosa* (50 and 150 mg/kg/day, groups 2 and 3, respectively) in the hole board assay and the locomotor activity test are shown in Figure 2. No significant differences were observed between groups 2 or 3 and group 1 (the control group) in the number of holes explored or in the locomotor activity, which indicates that *V. carnosa* did not induce sedation at these doses. However, the number of spatial explorations (rearings) was significatively increased in group 3, suggesting an anxiolytic-like behavior in mice at this dose. We have previously shown that chronic oral intake of hesperidin (50 and 100 mg/kg/day) produces anxiolytic-like effects in mice [83], and increases the number of rearings in the hole board assay. However, hesperidin doses were higher than its concentration in the *V. carnosa* extract, other components or interactions may be responsible for the extract effects. 

On the other hand, we have previously shown that a single intraperitoneal injection of aqueous 1 extract of *V. carnosa* (500 mg/kg) reduce locomotor activity and hole explorations in mice, producing a sedative effect. In contrast, a single oral administration (600 mg/kg) has shown anxiolytic-like effects [10]. All these properties of *V. carnosa* aqueous 1 extract as a CNS depressant are consistent with those previously reported for this species.

#### 2.7.4. Effect of *V. carnosa* in the Y-maze Test

In mice chronically treated with the aqueous 1 extract of *V. carnosa* (group 2 and 3) the alternation index was significantly higher than in those that drank water (group 1) (Figure 3). This means that mice in groups 2 and 3 were better able to remember Y-maze arms previously visited. Similarly, *V. carnosa* groups 2 and 3 showed a significantly lower return index than group 1, which also suggests that *V. carnosa*-treated mice had better working memory. This type of memory is one of the memories usually impaired in AD patients. Rearings were also measured in this assay (Figure 3). This exploratory parameter was also significantly increased in group 3, again showing the anxiolytic-like properties induced by this dose of *V. carnosa*. It is noteworthy that these two effects on memory and anxiety can be observed.

Other *Valeriana* species have shown effects on memory *in vivo*. For example, an ethanolic extract of *V. amurensis* (0.2–0.8 g/kg orally for 14 days) has improved cognitive function in the Morris water maze test in a mouse model of Aβ-induced cognitive dysfunction. This extract also improved brain cholinergic function by increasing ACh secretion, enhancing choline acetyltransferase activity, and protecting brain neurons from Aβ-induced apoptosis [35]. On the other hand, the roots of *Urtica dioica* (100 mg/kg orally for 19 days) attenuated scopolamine-induced neuropathic effects on rat learning and memory evaluated in the Y-maze assay. Moreover, this extract showed antioxidant properties and protected the brain from neuronal death [84]. 

#### 2.7.5. Effect of Chronic Treatment of *V. carnosa* on AChE Inhibition in the Mouse Brain 

AChE activity was significantly decreased in the brains of mice treated with the aqueous 1 extract of *V. carnosa* (groups 2 and 3) for 30 days compared with the control group (group 1) (Figure 4). This neurotransmitter is highly involved in cognitive functions of memory and learning [85,86] and mechanisms related to attention [87,88]. Consequently, inhibition of AChE activity in *V. carnosa*-treated mice may result in increased ACh availability, which may in turn have contributed to their improved performance in the Y-maze assay.

These results obtained for *V. carnosa* are promising and comparable to those obtained for other species reported in the literature. A significant decrease in AChE activity has been observed in the cortex and hippocampus of mice after administration of an *n*-hexane extract of the aerial parts of *Lycopodium thyoides* and *Lycopodium clavatum* (25, 10, and 1 mg/kg i.p., a single administration) [89]. Aqueous (400 mg/kg) and alkaloid-enriched (100 mg/kg, 200 mg/kg, and 400 mg/kg) extracts of *Erythrina velutina* leaves significantly inhibited cortical ChEs in mice after acute oral administration [90]. Moreover, a single intraperitoneal administration of linarin (35, 70, and 140 mg/kg), a flavone glycoside found in *V. officinalis* [9], decreased AChE activity in the cortex and hippocampus to an extent similar to that of huperzine A (0.5 mg/kg), the positive control compound [91].

#### 2.7.6. Effect of *V. carnosa* in the Tail Suspension Test

Immobility time in the tail suspension test was significantly decreased in mice treated with *V. carnosa* (groups 2 and 3) *vs.* group 1 (Figure 5), which indicates an antidepressant-like effect of this species. Although the mechanism of action underlying this effect remains to be thoroughly established, MAO inhibition may be thought to play a role. Other *Valeriana* species have shown antidepressant-like properties in rodents. For instance, methanolic, aqueous, and aqueous-ethanolic extracts of *V. wallichii* (acute 50 to 200 mg/kg) have shown antidepressant-like activity in the tail suspension test [31]. An extract of *V. glechomifolia* enriched in valepotriates produced a biphasic (1–20 mg/kg p.o.) dose-response relationship without changes in locomotor activity or motor coordination (assessed in the open field and rotarod tests, respectively) of the animals [92].

Worth pointing out, anxiety and depression are the most common neuropsychiatric disorders in AD [93]. Thus, extracts from *V. carnosa* may have a dual function in these patients. Moreover, the inhibition of MAOs may reduce oxidative stress in the brain by reducing H_2_O_2_ concentration.

#### 2.7.7. Effect of *V. carnosa* on Oxidative Stress Parameters

Oxidative stress plays an important role in the pathogenesis of AD. The brain is more susceptible to oxidative stress than other organs, and many neuronal components (lipids, proteins, and nucleic acids) can be oxidized in AD because of mitochondrial dysfunction, increased metal concentration, inflammation, and Aβ aggregation. Further, oxidative stress is involved in the development of AD by promoting Aβ deposition, tau hyperphosphorylation, and subsequent loss of synapses and neurons. These data suggests that oxidative stress is an essential component of the disease process so antioxidants may be useful treatment options [94]. 

Figure 6 shows that mice treated with 50 mg/kg/day *V. carnosa* (group 2) had lower TBARS concentration and higher GSH levels than control mice (group 1) (Figure 6a and Figure 6b, respectively). Because TBARS are formed as a byproduct of lipid peroxidation, chronic ingestion of *V. carnosa* extract may inhibit lipid peroxidation. This result was further enhanced by the presence of high GSH levels. GSH is one the first line defense of mitochondrial membranes against oxidative damage, as it ensures the reduction of hydroxy peroxide to phospholipids and other lipid peroxides. This property is not unique to *V. carnosa*, as many plant extracts have been described as antioxidants. However, this native valerian has distinctive multitarget properties which can be combined with the antioxidant effects of most medicinal plants. For example, *V. wallichii* powder has been reported to prevent the decrease in GSH levels and the increase in TBARS concentration caused by methylmercury in enriched mitochondrial fractions of adult Wistar rats [95]. Moreover, the tincture of *V. officinalis* prevents the oxidative damage induced by different prooxidants (quinolinic acid, sodium nitroprusside, 3-nitropropionic acid, Fe^2+^ and Fe^2+^/ethylenediaminetetraacetic acid (EDTA) complex) in rat brain homogenates [96]. Furthermore, an alcoholic extract of *V. officinalis* has been reported to protect hippocampal neurons from iron/ascorbate-induced lipoxidation and neurotoxicity in a model of Parkinson’s disease [97].

## 3. Materials and Methods

### 3.1. Chemicals and Reagents

All chemicals and drugs – DPPH, 6-hydroxy-2,5,7,8-tetramethylchromium-2-carboxylic acid (Trolox), ABTS, ferrozine, EDTA, 5,5′-dithiobis(2-nitrobenzoic acid) (DTNB), acetylthiocholine iodide and butyrylthiocholine iodide, reduced glutathione, gallic acid, tacrine, pargyline, clorgyline, isatin, *L*-deprenyl were purchased from Sigma-Aldrich®, Argentina. TBA, trichloroacetic acid (TCA) and butylated hydroxytoluene (BHT) were purchased from Biopack® Productos Químicos, Argentina. Vitamin E was purchased from Casasco Laboratories (Argentina).

Murine mAChE and recombinant hBChE were kindly provided by Xavier Brazzolotto and Florian Nachon (IRBA, Brétigny-sur-Orge, France). Recombinant human microsomal hMAO (expressed in baculovirus-infected insect cells (BTI-TN-5B1-4 cells)), horseradish peroxidase (type II, lyophilized powder) and *p*-tyramine hydrochloride were obtained from Sigma Aldrich (Sigma Aldrich, St. Louis, MO, USA). 10-Acetyl-3,7-dihydroxyphenoxazine (Amplex Red) was synthesized according to [98]. All reagents and solvents used were analytical or HPLC grade. 

### 3.2. High Performance Liquid Chromatography (HPLC)

Analytical HPLC fractionations were performed using an LKB Pharmacia instrument with C-18 reversed-phase Vydac columns (5 mm, 0.46 × 25 cm) (The Separation Group, Hesperia, CA, USA). The aqueous 1 extract of *V. carnosa* was injected into the column and eluted with an aqueous acetonitrile (ACN) gradient (*v*/*v*) (30 min total run time): 10% isocratic, 0–10 min, 10–45% ACN, 10–20 min; 45–80% ACN, 20–30 min. The elution flow was kept constant at 1 mL/min and 200 µL of each sample was injected. The effluent was monitored at 280 nm. All analyses (retention times) were performed in triplicate.

### 3.3. Animals

#### General

Adult male Swiss mice weighing 25–30 g obtained from the Central Vivarium of the School of Pharmacy and Biochemistry of the University of Buenos Aires were used throughout. For behavioral trials, mice were housed in groups of four to five animals in a controlled environment (20–23 °C), with free access to food and water, and maintained on a 12 h/12 h day/night cycle, switched on at 07:00 a.m. The number of animals used was the minimum number compatible with obtaining significant data. Mice were randomly assigned to one of the three groups and used only once. Pharmacological assays were performed by experimenters who were blinded to treatments and were carried out between 10:00 a.m. and 2:00 p.m. Protein concentration was determined by the Bradford method using bovine serum albumin as a standard [99].

### 3.4. Chemical Assays

#### 3.4.1. Determination of Antioxidant Capacity by the DPPH Method

The ability of extracts to scavenge free radicals was determined by the DPPH assay [98]. Briefly, 250 µL of 500 µM DPPH and 200 µL of 100 mM Tris HCl buffer, pH 7.4, were incubated with 50 µL of different concentrations of each extract for 20 min at room temperature in the dark. Absorbance was measured at 517 nm. Trolox was used as a positive control. Measurements were performed in triplicate.

#### 3.4.2. Determination of Antioxidant Capacity by the ABTS•+ Method

This procedure is based on the evaluation of the free radical scavenging capacity of the extracts to reduce the radical cation ABTS•+ to ABTS [100]. ABTS•+ was prepared by reacting 7 mM ABTS (in deionized water) with 2.45 mM K_2_S_2_O_8_. The solution was kept at room temperature in the dark for at least 12 h to obtain stable absorbance values at λ = 734 nm. Subsequently, the reagent was diluted to obtain an approximate absorbance of 0.7 (water as blank). A 50 µL aliquot of the different concentrations of each extract was added to 450 µL of this solution. The decrease in absorbance at 734 nm reflects the antioxidant activity. Trolox was used as a positive control (0 and 20 µM).

The percentage of scavenging activity was calculated as (for Section 3.4.1 and Section 3.4.2):(1)$$Free\;radical\;scavenging\;activity\;\left(\%\right)=\left(\frac{A_{control}-A_{sample}}{A_{control}}\right)\times \;100$$

*A_control_* = absorbance of the control prepared without extracts.

*A_sample_* = absorbance with the tested extract.

The 50 % of the effective concentration (EC_50_) was calculated as:(2)$$EC_{50}=antilog\;(D-\left[\left(A-50\%\max response\right)\times \left(D-C\right)\right]/\left(A-B\right)$$

*A*: immediate response over 50%.

*B*: immediate response below 50%.

*C*: log of the concentration corresponding to B.

*D*: log of the concentration corresponding to A.

#### 3.4.3. Metal Fe^2+^ Chelating Ability 

Metal iron has been identified within amyloid plaque nuclei in post-mortem brain samples of AD patients [101]. In addition, Fe^2+^ is one of the major chemical species involved in the initiation and propagation of ROS and lipid peroxidation [102]. Therefore, extracts capacity to chelate Fe^2+^ was determined using the ferrozine assay. For this study, reaction mixtures containing 200 µL of different concentrations of aqueous 1 extracts, 150 µL of 2 mM FeCl_2_ and 150 µL of 5 mM ferrozine were incubated at 37 °C for 10 min. Then, absorbance of the ferrozine-Fe^2+^ complex was then measured at 562 nm. Results were expressed as the percentage inhibition of the ferrozine-Fe^2+^ complex formation relative to the reactions without extracts, as follows:(3)$$Inhibition\;of\;ferrozine\;complex\;formation\;\left(\%\right)=\left(\frac{A_{control}-A_{sample}}{A_{control}}\right)\times \;100$$

*A_control_* = absorbance of control prepared without extracts.

*A_sample_* = absorbance with the extracts.

These data were used to calculate the concentration of extract required for 50% Fe^2+^ complex inhibition formation (IC_50_) [103]. EDTA was used as a positive control. IC_50_ values were determined by plotting the inhibition percentages against the extract concentrations applied using the GraphPad Prism8 software (GraphPad Software, San Diego, CA, USA).

### 3.5. Biochemical Assays

#### 3.5.1. Assessment of Lipid Damage through the Measurement of TBARS

The main substance reactive with TBA, MDA, is a pink dialdehyde formed as a by-product of the peroxidation of polyunsaturated fatty acids. The determination of MDA content in biological material is thus a convenient and widely used method for the quantitative assessment of lipid peroxidation [104]. 

Lipid damage was evaluated by measuring TBARS in two conditions:

***in vitro***: effect of all aqueous 1 extracts on brain homogenates of *naïve* mice.

***in vivo***: effect of aqueous 1 extract of *V. carnosa* on brain homogenates of mice after chronic ingestion (50 mg/kg/day) (group 2) (see Figure 7).

Brains from mice treated with *V. carnosa* (50 mg/kg/day in drinking water, group 2, *n* = 4), as well as the mice that drank water (control mice, group 1, *n* = 4) (see Figure 7), or brains from naïve mice (*n* = 3), were homogenized in 50 mM Tris HCl buffer (pH = 7.4) (1 g of brain tissue in 5 mL of buffer). Homogenates were centrifuged at 2500 rpm for 10 min and each supernatant was separated. For *in vitro* assays only, 0.2 mL of the supernatant from naïve mice (which were pooled) were incubated with 0.2 mL of the aqueous 1 extract of each plant or buffer (negative control) at 37 °C for 1 h. For both conditions (*in vitro* and *in vivo* TBARS assays), 0.04 mL of 4% BHT (in ethanol) and 0.4 mL of 20% TCA were added to precipitate the proteins and centrifuged at 4000 rpm for 10 min. Then, 0.4 mL of 0.7% TBA was added to the supernatant and refluxed for 1 h at 100 °C. At the end of this time, absorbance was measured at 535 nm [105]. Vitamin E was used as a positive control in the *in vitro* assay. The TBARS concentration was calculated as follows:(4)$$TBARS\;\left(nmol/g\;tissue\right)=\left(\frac{Abs}{156\frac{1}{mM\times \;cm}\times \;g\;tissue\;}\right)\times \;1000$$
(5)$$Inhibition\;of\;TBARS\;formation\;\left(\%\right)=\left(\frac{A_{control}-A_{sample}}{A_{control}}\right)\times \;100$$

*A_control_* = absorbance of control prepared with buffer.

*A_sample_* = absorbance with the extracts.

These data were used to calculate the IC_50_. IC_50_ values were obtained by plotting inhibition percentages against the extract concentrations applied using the GraphPad Prism8 software (GraphPad Software, San Diego, CA, USA).

#### 3.5.2. Determination of Reduced GSH

GSH levels in mouse brains were evaluated to estimate endogenous defenses against oxidative stress after chronic ingestion of *V. carnosa* (50 mg/kg/day, group 2, *n* = 4). The method was based on the reaction of Ellman’s reagent (DTNB) with free thiol groups. GSH production levels were determined as described by Sedlak and Lindsay [106]. Briefly, a calibration curve was made with GHS as a standard. The homogenate of each brain (1 g of brain tissue in 5 mL of 50 mM Tris HCl buffer, pH = 7.4) was deproteinized with 50% TCA and centrifuged at 3000 rpm for 10 min. This supernatant (0.2 mL) was added to 0.4 mL of 0.4 M Tris buffer (pH = 8.9) containing 0.02 M EDTA followed by the addition of 20 µL of 0.01 M DTNB solution in water. Then, the mixture was adjusted to pH = 8–9 with 0.2 M Tris buffer (pH = 8.2). Finally, absorbance was measured at 412 nm (Jasco V-550 spectrophotometer), and the results were expressed as µg of GSH/g of tissue.

#### 3.5.3. Inhibition of Aβ_1-42_ Aggregation

The thioflavin T (ThT) fluorometric assay was used to assess the ability of all aqueous 1 extracts to inhibit Aβ_1–42_ aggregation. Recombinant Aβ_1-42_ peptide was dissolved in DMSO (75 µM), and a dilution was made with 150 mM HEPES buffer (pH = 7.4, 150 mM NaCl; final DMSO concentration 10 %, *v*/*v*). Stock solutions of extracts and resveratrol (positive control) were prepared using DMSO. Reactions were carried out with final concentrations of 10 µM ThT, 1.5 µM Aβ_1–42_ solution, 0.1 mg/mL of each aqueous extract 1 or 10 µM resveratrol, and 3% (*v*/*v*) DMSO. ThT fluorescence was measured every 300 s (λex = 440 nm, λem = 490 nm) for 2 days, with the medium moving continuously between measurements on a plate reader (Synergy™ H4; BioTek Instruments, Inc., Winooski, VT, USA) at room temperature. Results were expressed as the percentage of inhibition, as follows:(6)$$\%\;inhibition=\left(1-\frac{F_{i}}{F_{0}}\right)\times \;100$$

*F_i_* = increase in fluorescence of Aβ_1–42_ treated with test extracts.

*F*_0_ = increased fluorescence of Aβ_1–42_ alone.

#### 3.5.4. ChE Inhibition

##### 3.5.4.1. Recombinant Enzymes

The inhibitory capacity of the extracts against mAChE and hBChE was determined by Ellman’s method [107]. Enzyme dilutions were made in 0.1 M sodium phosphate buffer (pH = 8.0). The stock solution of extracts was prepared in water:ethanol (1:1). Final reaction concentrations were: 370 μM DTNB, 500 μM substrate (acetylthiocholine or butyrylthiocholine iodide), and 100 pM mAChE or 1–2 nM hBChE. Absorbance at 412 nm was measured using a microplate reader (Synergy H4; BioTek Instruments, Inc., Santa Clara, CA, USA) during 1 min. Tacrine was used as a positive control.

The results were expressed as percentage of inhibition, as follows:(7)$$\%\;inhibition=100-\left(\frac{v_{i}}{v_{0}}\times \;100\right)$$

*v*_0_ = initial velocity calculated from the slope of the linear trend obtained without extracts.

*v_i_* = initial velocities in the presence of the extracts.

Each measurement was performed in triplicate. For IC_50_ measurements, different concentrations of extracts were used to obtain enzyme activities between 5% and 90%. IC_50_ values were obtained by plotting residual enzyme activities against inhibitor concentrations using GraphPad Prism 8 software (GraphPad Software, San Diego, CA, USA).

##### 3.5.4.2. AChE Inhibition in Mouse Brains

Brains from naïve mice (*n* = 3 and pooled), brains from *V. carnosa*-treated mice (group 2, *n* = 4; Group 3, *n* = 5), and brains of mice that had drunk water (control mice, group 1, *n* = 5) (see Figure 7) were homogenized in 50 mM Tris HCl buffer (pH = 7.4). The homogenates were then centrifuged at 5000 rpm at 4 °C for 10 min. Reactions were performed in a final volume of 500 µL of a 50 mM Tris HCl (pH = 7.4) solution, containing 310 µM DTNB, 600 µM acetylthiocholine, and a dilution of brain homogenate (the protein concentration of the incubation mixture was 0.135 µg/µL). For the *in vitro* assay (effect of aqueous 1 extracts on brain homogenates of naïve mice), the brain homogenate and extracts were pre-incubated at 37 °C for 5 min, to allow complete equilibration of the enzyme-inhibitor complexes. For the *in vivo* assay (effect after chronic ingestion of *V. carnosa* aqueous 1 extract on mouse brain homogenates), each homogenate was pre-incubated with buffer. The reaction was monitored for 10 min as a change in absorbance at 412 nm (Shimadzu Corporation™ UV-160A UV-VIS spectrometer, Analytical Instruments Division, Kyoto, Japan).

##### 3.5.4.3. BChE Inhibition in Mouse Serum 

Blood samples from naïve animals (*n*= 3) were collected by retroorbital puncture, and plasma was then isolated by centrifugation at 5000 rpm for 5 min. These reactions were performed in a final volume of 500 µL of a 50 mM TrisHCl solution (pH = 7.4), containing 310 µM DTNB, 200 µM butyrylthiocholine iodide and a 1/600 dilution of mouse serum as BChE source. The serum dilution and extracts were preincubated at 37 °C for 5 min, to allow complete equilibration of the enzyme-inhibitor complexes. Reactions were started by adding the butyrylthiocholine iodide and DTNB at room temperature. The reaction was measured for 5 min as a change in absorbance at 412 nm (Shimadzu Corporation™ UV-160A UV-VIS spectrometer, Analytical Instruments Division, Kyoto, Japan). Each concentration was measured at least in triplicate.

The inhibitory capacity of extracts (Section 3.5.4.2 and Section 3.5.4.3) was expressed as percent inhibition and calculated as described in Section 3.5.4.1. For IC_50_ measurements, different extract concentrations were used to obtain enzyme activities ranging from 5% to 90%. IC_50_ values were calculated by plotting the percent inhibition as a function of the logarithm of the inhibitor concentration applied, with the experimental data fitted to the four-parameter Hill equation using GraphPad Prism 8 software (GraphPad Software, San Diego, CA, USA).

#### 3.5.5. *In vitro* hMAO Inhibition Assay

The ability of the extracts to inhibit hMAO was determined according to Košak and coworkers [108]. Each extract was prepared as a solution so that the final concentration in the reaction tube was 1 mg/mL. Briefly, 100 μL of 50 mM sodium phosphate buffer, pH 7.4, 0.05% (*v*/*v*) Triton X-114 containing either extracts or reference inhibitors (pargylin, clorgyline, isatin, and *L*-deprenyl) and hMAO-A or hMAO-B were incubated for 15 min at 37 °C in black 96-well flat-bottomed microplates placed. After 15 min of preincubation, the reaction was started by adding *p*-tyramine, Amplex Red, and horseradish peroxidase (HRP) (1 mM, 250 μM, and 2 U/mL, respectively, in a final volume of 200 μL) and different extract concentrations or reference inhibitors. Resorufin fluorescent emission was quantified (λex = 530 nm, λem = 590 nm) at 37 °C, for 30 min where fluorescence increased linearly. Ethanol:water (1:1) was used as a control. To determine the blank (*b*), the enzyme solution was replaced by buffer solution. Initial velocities were calculated from the trends obtained, with each measurement performed in duplicate. Extract ability to inhibit MAOs was expressed as the percentage of inhibition according to the following equation:(8)$$Inhibition\;\left(\%\right)=100-\frac{V_{i}-b}{V_{o}-b}\times \;100$$

*V_i_* = velocity in the presence of the extracts or reference compounds.

*V_o_* = control velocity in the presence of ethanol:water 1:1.

IC_50_ was calculated by plotting percent inhibition against inhibitor concentrations, with experimental data fitted to a four-parameter Hill equation using GraphPad Prism 8.0 (GraphPad Software, San Diego, CA, USA).

Extracts showed no interference with the reagents in the medium of the assay.

### 3.6. Pharmacological Studies

#### 3.6.1. Protocol

Animals had access to liquid and food *ad libitum* and were divided into three groups according to the corresponding oral treatment:

**Group 1:** Control (water, *n* = 14);

**Group 2:***Valeriana carnosa* total aqueous remaining phase, aqueous 1 (50 mg/kg/day, *n* = 12);

**Group 3:***Valeriana carnosa* total aqueous remaining phase, aqueous 1 (150 mg/kg/day, *n* = 12).

The dry fractions of aqueous 1 extracts of *V. carnosa* were dissolved in water in an amount sufficient for each mouse to consume the doses indicated. Extract concentration in the liquid was determined from the average of daily fluid intake (mL/mouse/day) and the average of body weight per mouse (g/mouse) to achieve the desired doses (50 and 150 mg/kg/day). The doses reported in this study (50 and 150 mg/kg/day) were obtained by calculating these results. Body weight was checked every 3 to 4 days and fluid intake was determined daily, as approximately 200 mL/kg, and no significant differences were found between the experimental groups. Extract solutions were replaced every 2 days. Stability controls of the extract was performed by HPLC. The extracts doses used in this study were chosen on the basis of pilot experiments and previous work, so that they had no effect on the locomotor activity of the animals [9,10,11,109]. 

#### 3.6.2. Behavioral Studies

*In vivo* studies were performed according to the protocol shown in Figure 7. The experimental apparatus was cleaned with 60% ethanol before each test. Animals were used only once. All experiments were filmed for subsequent analysis. Animals were randomly selected from each group to perform the biochemical tests.

##### Hole Board Assay

Mouse behavior was evaluated in a black plexiglass box with a 60 cm × 60 cm floor and 30 cm high walls, with four evenly spaced and centered holes in the floor, each 2 cm in diameter, as previously described [110] and illuminated with indirect dim light. Each animal was placed in the center of the field and allowed to move freely in the apparatus for 5 min. The following parameters were recorded: the number of times the mouse put its head into the hole (holes) and the number of vertical explorations, defined by animal standing on its hind legs (rearings) and leaving its front legs free or resting against the wall. The test was performed on day 15 of treatment (Figure 7). Results were expressed as relative (in %) to group 1 (control), which ingested water.

##### Locomotor Activity Test

Spontaneous locomotor activity was studied in a 15 cm × 30 cm transparent plexiglass box with 15 cm high walls. Each mouse was placed in the center of the box and allowed to explore freely for 5 min. The total distance traveled by each mouse was measured and expressed as total distance traveled relative (in %) to group 1. This test was performed, for each mouse, immediately after the hole board assay.

##### Y-Maze

The Y-maze was used to assess spatial working memory, as previously described [111] using an apparatus consisting of three arms of equal length (30 cm × 7 cm × 15 cm). Each mouse was placed at the end of one arm and allowed to freely explore the maze for 8 min. The entry sequence of each mouse into the different arms was documented to calculate the index of alternation and returns, as follows:(9)$$Alternation\;index=\frac{Alternations}{Total\;number\;of\;entries-2}*100$$
(10)$$Return\;index=\frac{Returns}{Total\;number\;of\;entries-2}*100$$

A spontaneous alternation occurs when a mouse enters a different arm of the maze at each of 3 successive arm entrances. A return, on the other hand, occurs when a mouse returns to the same branch from which it came. The number of rearings was also recorded, as a measure of exploration. The test was performed on day 22 of treatment (Figure 7).

##### Tail Suspension Test

The total duration of immobility induced by tail suspension was measured as previously described [112]. This is a behavioral test for mice suitable for detecting potential antidepressant drugs [113]. Mice were individually suspended by the tail from a metal hook (distance from the floor: 18 cm) with an adhesive tape (distance from the tip of the tail: 2 cm) for 6 min. Initially, mice try to escape the inverted position, but after a while they remain immobile, which is considered a sign of learned hopelessness. The duration of immobility was recorded during the last 4-min interval of the test. The test was performed on day 28 of treatment (Figure 7).

### 3.7. Statistical Analysis

Data from behavioral experiments were expressed as mean ± standard error of the mean (SEM). Values that were more than twice their standard deviation away from the mean were excluded from the statistical analysis. Data were analyzed by unpaired *t*-test or one-way analysis of variance (ANOVA), and post hoc comparisons between treatments and control were performed using Dunnett’s multiple comparison (GraphPad Prism 8.0, GraphPad Software, San Diego, CA, USA). Significance level was set at *p* < 0.05.

## 4. Conclusions

We have described the effects of *Valeriana effusa*, *V. ferax*, *V. macrorhiza*, *V. clarionifolia*, *V. carnosa*, and *V. officinalis* on various pharmacological targets associated with AD. All extracts were found to inhibit ChEs, especially BChE. Most of them also inhibited MAOs, and *V. carnosa* was the most potent inhibitor of hMAO-A. In most cases, the ethylic extracts exerted a broader activity profile than corresponding aqueous extracts. Furthermore, all aqueous 1 extracts inhibited Aβ aggregation, were able to scavenge free radicals, chelated Fe^2+^, and inhibited lipid peroxidation *in vitro*. Chronic treatment with aqueous 1 extract of *V. carnosa* improved mouse working memory and induced antidepressant and anxiolytic-like activities. Moreover, a decrease in AChE activity and a decrease in oxidative stress parameters were observed in mouse brains. Sleep-inducing properties have already been described for extracts of *V. carnosa*. 

All these properties suggest that *Valeriana carnosa* can be a multifunctional phytotherapeutic alternative for the treatment of AD and its comorbidities (such as anxiety, depression, and sleep disorders). Although preclinical and clinical trial phases still need to be completed, ancient widespread use of this medicinal plant suggests it may present fewer side effects than currently used drugs and thus gain faster market access. Moreover, it could also avoid polypharmacotherapy typically prescribed to the elderly and its consequent adverse effects.

## Figures and Tables

**Figure 1 pharmaceuticals-16-00129-f001:**
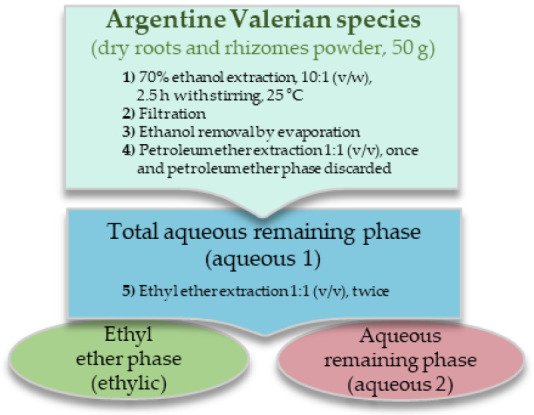
Extraction scheme of the *Valeriana* species roots and rhizomes.

**Figure 2 pharmaceuticals-16-00129-f002:**
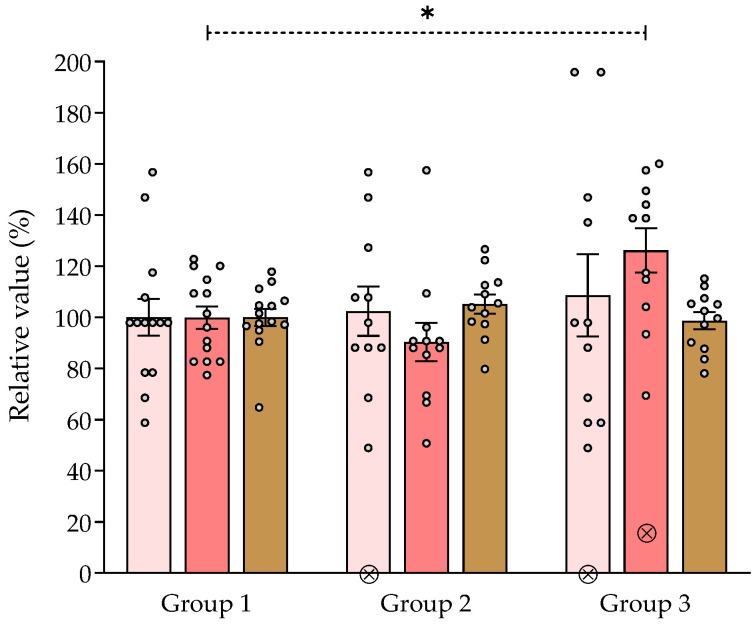
Effect of *Valeriana carnosa* aqueous 1 extract (on day 15 of treatment) in the hole board and locomotor activity assays (50 and 150 mg/kg/day in drinking water, groups 2 and 3, respectively). Results are expressed as mean ± SEM. of percent exploring holes (
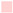
), percent spatial exploration (rearings) (
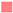
) and percent distance traveled in the locomotor activity assay (
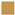
) recorded in 5-min sessions. * *p* < 0.05, significantly different from group 1. Dunnett’s multiple comparison test after ANOVA. ⊗: discarded values. n_group 1_ = 14, n_group 2_ = 12, n_group 3_ = 12.

**Figure 3 pharmaceuticals-16-00129-f003:**
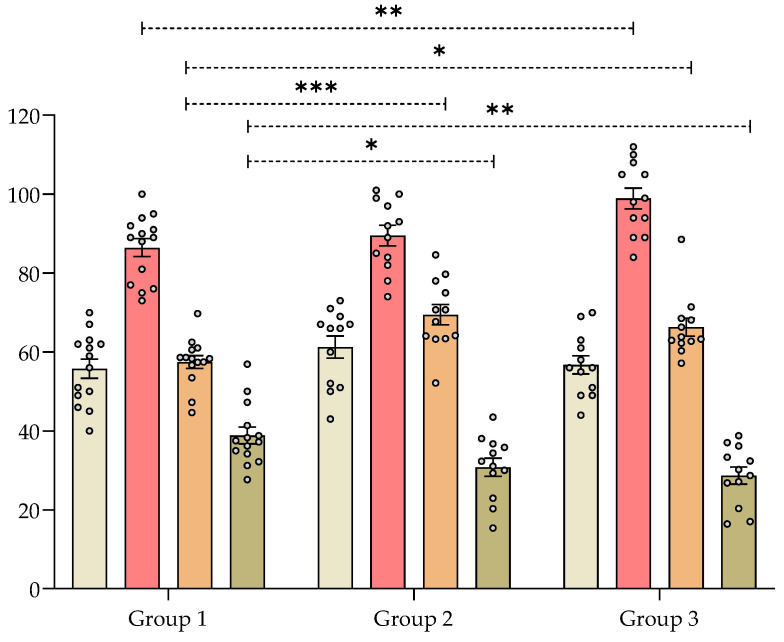
Effect of *Valeriana carnosa* aqueous 1 extract (50 and 150 mg/kg/day in drinking water, groups 2 and 3, respectively; on day 22 of treatment) in a Y-maze test in mice. Results are expressed as mean ± SEM of total number of entries (
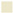
), number of rearings (
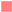
), index of alternations (
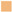
) and returns (
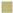
) recorded in 8-min sessions. * *p* < 0.05, ** *p* < 0.008, *** *p* < 0.005 significantly different from group 1 (control). Dunnett’s multiple comparison test after ANOVA. n_group 1_ = 14, n_group 2_ = 12, n_group 3_ = 12.

**Figure 4 pharmaceuticals-16-00129-f004:**
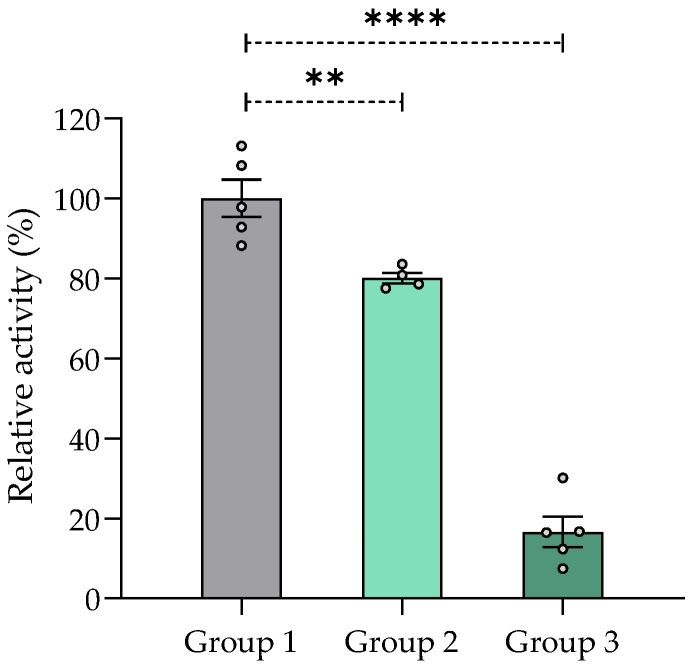
AChE inhibition (%) in mouse brain homogenates after chronic treatment with aqueous 1 extract of *Valeriana carnosa* (50 and 150 mg/kg/day in drinking water, groups 2 and 3, respectively). Results are expressed as mean ± SEM of AChE activity (Abs/min) in % relative to control group 1 (mice drinking water); ** *p* < 0.01, **** *p* < 0.0001, significantly different from group 1; Dunnett’s multiple comparison test after ANOVA. n_group 1_ = 5, n_group 2_ = 4, n_group 3_ = 5.

**Figure 5 pharmaceuticals-16-00129-f005:**
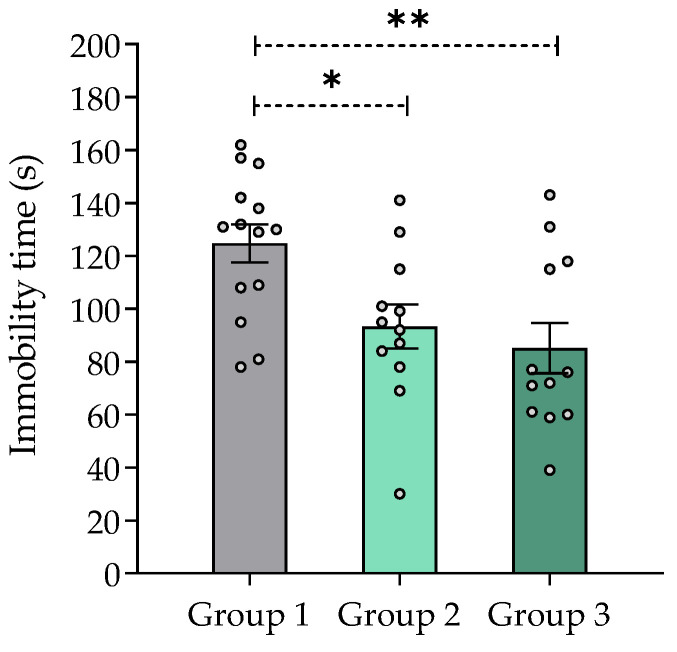
Effect of *Valeriana carnosa* aqueous 1 extract (50 and 150 mg/kg/day in drinking water, groups 2 and 3, respectively, on day 28 of treatment) in the tail suspension test in mice. Results are expressed as mean ± SEM, recorded in 6 min sessions * *p* < 0.05, ** *p* < 0.008, significantly different from control (group 1). Dunnett’s multiple comparison test after ANOVA. n_group 1_ = 14, n_group 2_ = 12, n_group 3_ = 12.

**Figure 6 pharmaceuticals-16-00129-f006:**
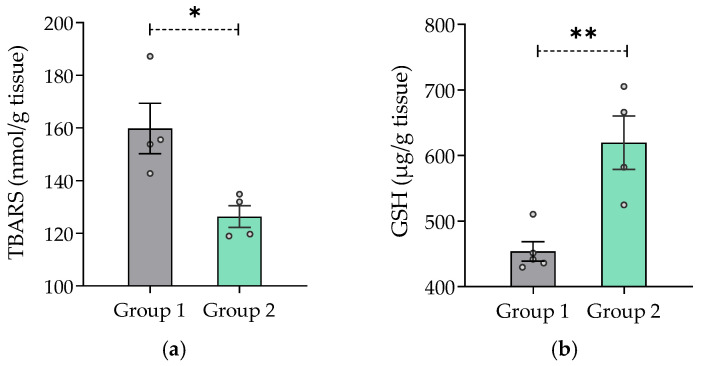
*In vivo* antioxidant capacity of *Valeriana carnosa* aqueous 1 extract. (**a**) TBARS concentration expressed as nmol/g of tissue (n_group 1_ = 4, n_group 2_ = 4) and (**b**) GSH concentration expressed as µg/g of tissue (n_group 1_ = 5, n_group 2_ = 4) present in mouse brain homogenates that have ingested 50 mg/kg/day of *Valeriana carnosa* aqueous 1 extract in drinking water for 30 days (group 2). * *p* < 0.05, ** *p* < 0.005, significantly different from group 1 (control); unpaired *t*-test.

**Figure 7 pharmaceuticals-16-00129-f007:**
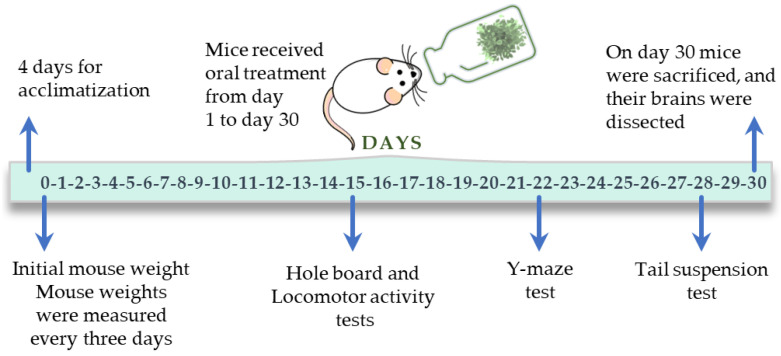
Experimental timeline followed in the *in vivo* study with *V. carnosa*.

**Table 1 pharmaceuticals-16-00129-t001:** Antioxidant capacity of *Valeriana* species aqueous extracts 1.

Sample	Total Phenols ^1^	DPPH	ABTS	Ferrozine	TBARS
	mg Galic Acid/ 1 g Plant	EC_50_ (mg/mL)	EC_50_ (mg/mL)	IC_50_ (mg/mL)	IC_50_ (mg/mL)
*V. effusa*	125.52 ± 6.80	0.56 ± 0.04	0.41 ± 0.02	2.59 ± 0.08	25.04 ± 0.79 ^2^
*V. ferax*	191.56 ± 8.99	0.57 ± 0.02	0.21 ± 0.13	2.10 ± 0.04	5.00 (2.38–10.51) ^3^
*V. macrorhiza*	252.85 ± 12.36	0.43 ± 0.02	0.13 ± 0.01	5.13 ± 0.09	0.20 (0.13–0.30) ^3^
*V. clarionifolia*	257.36 ± 19.55	0.30 ± 0.01	0.10 ± 0.01	1.3 ± 0.11	1.29 (0.81–2.05) ^3^
*V. carnosa*	890.25 ± 156.05	0.12 ± 0.02	0.04 ± 0.01	1.71 ± 0.10	0.18 (0.11–0.30) ^3,4^
*V. officinalis*	736.19 ± 34.27	0.32 ± 0.01	0.51 ± 0.06	4.76 ± 0.28	1.76 (1.25–2.48) ^3^
Trolox ^5^	ND	0.016 ± 0.006	0.013 ± 0.004	ND	ND
EDTA ^5^	ND	ND	ND	0.006 ± 0.001	ND
Vitamin E ^5^	ND	ND	ND	ND	37.73 ± 0.41 ^2^

All values are expressed as mean ± SEM of three independent determinations. ND: Not determined; ^1^ Data obtained from [10]; ^2^ Inhibition (%) at 1 mg/mL; ^3^ IC_50_: mean (confidence interval) of three independent determinations; ^4^ Data from *V. carnosa* ethylic extract; ^5^ Positive controls: trolox (DPPH and ABTS), EDTA (Ferrozine), vitamin E (TBARS).

**Table 2 pharmaceuticals-16-00129-t002:** Inhibition of AChE and BChE by *Valeriana* species extracts.

Sample	Extract	AChE IC_50_ (mg/mL) ^1^	mAChE Inhibition (%) ^2^	BChE IC_50_ (mg/mL) ^1^	hBChE Inhibition (%) ^2^
		Mice Brain	Recombinant	Murine Plasma	Recombinant
*V. effusa*	Aqueous 1	12.69 (5.8–27.7)	NA	0.86 (0.61–1.22)	17.2 ± 3.1
	Aqueous 2	ND	ND	0.65 (0.45–0.92)	ND
	Ethylic	ND	23.5 ± 3.4	0.41 (0.28–0.62)	37.3 ± 3.1
*V. ferax*	Aqueous 1	5.45 (3.69–8.05)	ND	0.53 (0.43–0.65)	18.7 ± 3.1
	Aqueous 2	ND	ND	0.15 (0.11–0.2)	ND
	Ethylic	ND	31.6 ± 4.1	0.025 (0.017–0.038)	69.5 ± 7.2
*V. macrorhiza*	Aqueous 1	1.08 (0.49–2.37)	NA	0.082 (0.07–0.10)	24.4 ± 2.4
	Aqueous 2	ND	ND	0.95 (0.57–1.58)	ND
	Ethylic	ND	29.9 ± 5.3	0.045 (0.004–0.005)	78.7 ± 3.9
*V. clarionifolia*	Aqueous 1	1.29 (0.81–2.05)	26.4 ± 2.7	0.0019 (0.0014–0.0024)	64.3 ± 1.1 IC_50_ (mg/mL): 0.190 (0.120–0.300)
	Aqueous 2	ND	ND	0.039 (0.030–0.050)	ND
	Ethylic	ND	32.2 ± 4.4	0.00057 (0.00041–0.00081)	67.9 ± 9.4
*V. carnosa*	Aqueous 1	6.71 (2.86–15.77)	NA	1.46 (0.99–1.14)	NA
	Aqueous 2	ND	ND	2.57 (1.11–6.0)	ND
	Ethylic	6.92 (2.14–22.41)	37.7 ± 5.8	0.26 (0.16–0.42)	52.6 ± 9.5
*V. officinalis*	Aqueous 1	3.42 (1.11–10.6)	NA	0.15 (0.13–0.18)	88.7 ± 1.4 IC_50_ (mg/mL): 0.140 (0.110–0.180)
	Aqueous 2	ND	ND	0.50 (0.33–0.75)	ND
	Ethylic	ND	ND	0.0041 (0.0026–0.0067)	ND
Tacrine ^3^		ND	IC_50_ (µM): 0.140 ± 0.008	ND	IC_50_ (µM): 0.023 ± 0.003

^1^ IC_50_: mean (confidence interval) of three independent determinations; ^2^ % Enzyme inhibition: mean ± SEM of three independent determinations. Aqueous 1 extracts evaluated at 1 mg/mL and ethylic extracts evaluated at 0.5 mg/mL; ^3^ Positive control: tacrine (mAChE and hBChE). NA: Not active; ND: Not determined.

**Table 3 pharmaceuticals-16-00129-t003:** Inhibition of MAO-A and MAO-B by *Valeriana* species extracts.

Sample	Extract ^1^	hMAO-A ^2^	hMAO-B ^2^
		Inhibition (%)	Inhibition (%)
*V. effusa*	Aqueous 1	37.2 ± 1.1	23.5 ± 1.3
	Ethylic	45.3 ± 0.9	14.2 ± 4.9
*V. ferax*	Aqueous 1	38.7 ± 0.1	22.1 ± 0.8
	Ethylic	57.9 ± 0.7	NA
*V. macrorhiza*	Aqueous 1	24.7 ± 3.7	62.7 ± 2.5
	Ethylic	68.6 ± 0.1	50.6 ± 3.6
*V. clarionifolia*	Aqueous 1	39.5 ± 0.6	39.9 ± 1.8
	Ethylic	0.4 ± 0.7	NA
*V. carnosa*	Aqueous 1	41.6 ± 0.1	57.6 ± 5.4
	Ethylic	69.5 ± 1.4 IC_50_ (mg/mL): 0.286 (0.213–0.384) ^3^	49.4 ± 4.2
*V. officinalis*	Aqueous 1	62.8 ± 0.4	57.5 ± 3.2
	Ethylic	ND	ND
Clorgyline ^4^	-	IC_50_ (µM): 0.00335 ± 0.00031 ^3^	IC_50_ (µM): 13.568 ± 1.157 ^3^
Pargyline ^4^	-	IC_50_ (µM): 3.968 ± 0.275 ^3^	IC_50_ (µM): 0.195 ± 0.019 ^3^
*L*-deprenyl ^4^	-	IC_50_ (µM): 62.664 ± 0.411 ^3^	IC_50_ (µM): 0.012 ± 0.004 ^3^
Isatin ^4^	-	ND	IC_50_ (µM): 19.778 ± 1.108 ^3^

^1^ Aqueous 1 extracts evaluated at 1 mg/mL and ethylic extracts evaluated at 0.5 mg/mL; ^2^ % hMAO inhibition: mean ± SEM of three independent determinations; ^3^ IC_50_: mean (confidence interval) of three independent determinations; ^4^ Positive controls: clorgyline (MAO-A), pargyline (MAO-B), *L*-deprenyl (MAO-B), isatin (MAO-B)NA: Not active; ND: Not determined.

**Table 4 pharmaceuticals-16-00129-t004:** Inhibition of Aβ_1–42_ aggregation by *Valeriana* species aqueous 1 extracts.

Sample	Aβ_1–42_ Aggregation ^1^
	Inhibition (%)
*V. effusa*	93.4% ± 4.1
*V. ferax*	47.8% ± 13.4
*V. macrorhiza*	32.0% ± 11.5
*V. clarionifolia*	81.6% ± 13.6
*V. carnosa*	65.7% ±12.2
*V. officinalis*	59.7% ± 2.6
Resveratrol ^3^	93.9% ± 4.8 ^2^

^1^ % inhibition of Aβ_1-42_ peptide aggregation at 0.1 mg/mL: mean ± SEM of three independent determinations. Four replicates each measurement; ^2^ Value of % inhibition at 10 μM; ^3^ Positive control: Resveratrol.

## Data Availability

Data are contained within the article and Appendix A.

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
