# Peer review of "Biological Evaluation of Valeriana Extracts from Argentina with Potent Cholinesterase Inhibition for the Treatment of Neurodegenerative Disorders and Their Comorbidities—The Case of Valeriana carnosa Sm. (Caprifoliaceae) Studied in Mice"

_pharmaceuticals, 2023, doi:10.3390/ph16010129_

Round 1

Reviewer 1 Report

In this manuscript, the authors evaluated in vitro and in vivo cholinesterase inhibition activity of Valeriana extract which was taken from Argentina. Some errors exist in this manuscript which are as follows:

 1.      The authors should take care of the language and polish the English language.

2.       IC50 values of AChE and BChE which are mentioned in the abstract and results and discussion part should match and discussed properly.

3.      Remove reference [1] from line 51. It is enough in line 53.

4.      Referencing should be updated. Add the latest references related to the research work.

5.      Mention/explain antioxidant results in the text also as mentioned in Table 1.

6.      Elaborate Table 2 results in comparison to literature reports.

7.      Add suitable references for MAO-A and MAO-B inhibitors (Lines 268-278).

8.      The challenges and future scope of the selected topic of the present manuscript may be included in the conclusions.

9.     There is a need to check reference style. References must be arranged on single format.

Reviewer 2 Report

This is a comprehensive analysis of the Valeriana extract's characteristics. However, there need some revisions as follows:

1- The exact number of mice in each group should be mentioned.

2- For in vitro analysis, there is no control, for example, the extraction buffer only, or a negative control

3- For analysis of percent values, Chi-square is better than ANOVA. 

4- The results need to be more organized for a better understanding for readers 

5- It would be ideal if authors could use the stress-exposed mice to more accurately evaluate the positive effects of the extracts

Reviewer 3 Report

Dear Authors,

The MS entitled " Biological evaluation of Valeriana extracts from Argentina with potent cholinesterase inhibition for the treatment of neuro-degenerative disorders and their comorbidities. The Case of Valeriana carnosa Sm. (Caprifoliaceae) studied in mice" was thoroughly reviewed. The manuscript is well composed and according to the format of the journal.  The authors have targeted Alzheimer's disease (AD) probable treatment using various probable causes of this disease i.e. β-amyloid plaques, oxidative stress, or alterations in neurotransmitter. A comparative study of the effect of extracts of valerians plants on different AD-related biological targets has been presented. The study is also accompanied in vivo. In my suggestion, the MS is very well designed. 

I suggest some minor corrections.

1.Check the font of all the equations used. Also corrected the tables and figures fonts. These do not rhyme with the overall MS.

2. What was the phytochemical profile of these plant extracts? Why did not the authors characterize these extracts? The results could be clearer if the phytochemicals nature of these extracts is known. GC-MS might be sufficient.

Add some recent/relevant literature in the introduction or where applicable related to anticholinesterase potential.

Hussain, H., Ahmad, S., Shah, S. W. A., Ghias, M., Ullah, A., Rahman, S. U., ... & Alghamdi, S. (2021). Neuroprotective Potential of Synthetic Mono-Carbonyl Curcumin Analogs Assessed by Molecular Docking Studies. Molecules26(23), 7168.

Khan, F. A., Khan, S., Khan, N. M., Khan, H., Khan, S., Ahmad, S., ... & Aziz, R. (2021). Antimicrobial and Antioxidant Role of the Aerial Parts of Aconitum violaceum. Journal of the Mexican Chemical Society65(1), 84-93.

Khan, S., Khan, H. U., Khan, F. A., Shah, A., Wadood, A., Ahmad, S., ... & Kamran, N. (2022). Anti-Alzheimer and Antioxidant Effects of Nelumbo nucifera L. Alkaloids, Nuciferine and Norcoclaurine in Alloxan-Induced Diabetic Albino Rats. Pharmaceuticals15(10), 1205.

Khan, F. A., Khan, N. M., Ahmad, S., Aziz, R., Ullah, I., Almehmadi, M., ... & Aljuaid, A. (2022). Phytochemical profiling, antioxidant, antimicrobial and cholinesterase inhibitory effects of essential oils isolated from the leaves of Artemisia scoparia and Artemisia absinthium. Pharmaceuticals15(10), 1221.

Round 2

Reviewer 1 Report

The revised version can now be accepted for publication.

Reviewer 2 Report

I have no more comments